# Seeing Through Language: How Text Reveals Object and State Bias in VLMs

## Abstract

Vision-Language models (VLMs) have demonstrated strong performance across a variety of multimodal benchmarks though not without internal biases. Little is known about how VLMs balance sensitivity to object identity versus object state. In this work, we systematically investigate object-state bias in VLMs by evaluating a broad set of models spanning diverse architectures and sizes. To enable controlled analysis, we introduce the Benchmark for Biases in Objects and States (BBiOS) dataset containing objects in both their original and transformed states. Across a variety of experiments, we examine model performance on recognizing objects, states, and their interactions. Our results reveal a consistent object bias, where models reliably recognize object categories but struggle to accurately capture states. Furthermore, attempts to steer models toward greater state sensitivity through prompting or injecting oracle information yield only marginal improvements. These findings highlight a fundamental limitation in current VLMs, suggesting that different training strategies or architectural innovations are required to reduce object-state bias in multimodal reasoning.

## 1 Introduction

Vision-Language models (VLMs) have achieved remarkable performance across a range of multimodal tasks (Dai et al., 2023; Li et al., 2024b), including image captioning(Alayrac et al., 2022; Mokady et al., 2021) and visual question answering(Li et al., 2023; Liu et al., 2023). These models leverage large-scale paired image-text datasets to learn a joint representation of visual and linguistic concepts. However, despite their success, a growing body of work shows that VLMs inherit and often amplify biases embedded in their training data(Huang et al., 2025; Zhou et al., 2022; Hirota et al., 2022; Hazirbas et al., 2024; Srinivasan & Bisk, 2022).

Biases in VLMs manifest in different ways. On the one hand, models tend to reinforce social and cultural stereotypes in their generated captions, for example, by associating certain roles or activities with specific genders or cultures (Hamidieh et al., 2024; Hirota et al., 2023). On the other hand, VLMs display a tendency to prioritize frequent categories while neglecting rare or nuanced occurrences (Parashar et al., 2024; Wang et al., 2024b; Shi et al., 2024a). Text attribution plays an important role in this problem, because captions or textual labels in training datasets are used as the primary signal for linking images and language. The way objects, actions, or states are described affects the model's semantic understanding (Zhao et al., 2024; Li et al., 2025). Captions may oversimplify complex visual phenomena or omit important context resulting in different model behaviours (Ye et al., 2025; Dong et al., 2024). These limitations not only affect the model's perfor-

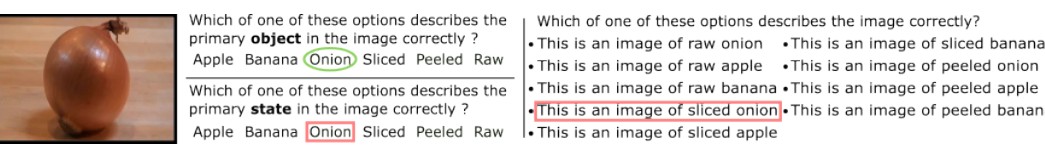

**Figure 1:** We investigate how VLMs are biased towards objects across different experimental setups using a new dataset benchmark. VLMs consistently achieve higher object accuracies than state accuracies.

mance but also limit generalization which can have detrimental effects on downstream tasks (Segalis et al., 2023; Shi et al., 2024b).

Object State Change (OSC), which is vital for a wide range of applications such as activity recognition and robotic manipulation is becoming a more common task. Traditionally, models focused on a limited set of known state changes within a predefined vocabulary, which constrains their effectiveness in real-world scenarios (Alayrac et al., 2017; Aboubakr et al., 2019). Recent efforts aim to develop more flexible and generalized approaches capable of identifying object state changes in open and unconstrained environments in order to increase robustness of such systems (Xue et al., 2024; Pan et al., 2025).

Despite these advances, VLMs' ability to handle object states remains limited. In a recent study Newman et al. (2024) introduce the ChangeIt-Frames dataset to test whether open-source VLMs encode the state of an object (e.g. a whole apple vs sliced apple). The results show that while these models perform reliably for object recognition, they consistently fail to identify objects' states. Another recent study, Kawaharazuka et al. (2024) further illustrates the challenge in real-world applications. Instead of treating states as discrete categories, this study investigates continuous changes such as butter melting and onions frying. The authors show that without additional optimization, VLMs often misinterpret these states.

This paper explores object and state bias in VLMs to understand how different models are effected. Figure 1 summarizes the framework used to examine object and state bias. A new benchmark is created to investigate this phenomenon consisting of images of kitchen ingredients in different states. We specifically focus on kitchen ingredients because their states are often visually distinct allowing for objective evaluation of the object-state bias in addition to having a many-to-many relationship between objects and states, i.e. potatoes and carrots can both be peeled and sliced. By systematically examining how models predict ingredients and their states, we highlight the gap in current VLMs and investigate how these biases may limit the performance for downstream tasks which require fine-grained reasoning.

Our contributions are as follows: (i) We introduce the Benchmarking Bias in Object State (BBiOS) dataset for evaluating object and state bias, the first of its kind. (ii) We present a framework for measuring the object and state biases across various VLM architectures. (iii) We evaluate the impact of these biases on the downstream task of visual reasoning across 24 VLMs. (iv) We show that steering and injected oracle knowledge does not solve the object bias, demonstrating inherent representation/training data issue.

## 2 RELATED WORK

### 2.1 SOCIAL BIASES

Recent studies show that Vision-Language Models (VLMs) trained on large web data inherit and amplify social biases. Ruggeri et al. (2023) provide a multi-dimensional bias analysis (gender, ethnicity, age) of VLMs and find that pre-trained models frequently produce stereotypical outputs. For example, when prompted with neutral image-based templates, VLMs produced derogatory continuations approximately 5% of the time, with a disproportionate focus on images depicting women and young people. Similarly Baherwani & Vincent (2024) shows that CLIP's embeddings of face images encode gender and racial stereotype: e.g., CLIP more often predicts the trait "smart" for images of Indian men than for others. Hausladen et al. (2025) reports that CLIP's social perception scores for faces are strongly affected by a person's age, gender, and race, with especially extreme values for images of black women. Girrbach et al. (2025) finds that VLMs such as LLaVA and InternVL display gender and occupational associations where, depending on the occupation, more positive skills and traits are attributed to women and more negative traits to men. The VisoGender benchmark (Hall et al. (2023) similarly finds that state-of-the-art VLMs show significant gender bias when resolving pronouns or occupations from images.

### 2.2 SHAPE AND TEXTURE BIAS

Bias in visual representations is not limited to social or cultural biases, but also appears in how models weigh different visual cues. A useful analogy for understanding object-state bias is the

longstanding study of shape vs texture bias in vision models. Just as object-state bias reflects the tendency of models to overemphasize object identity while under-representing object state, shape-texture bias reflects a preference of one visual cue over another. (Geirhos et al. (2018)) showed that standard CNNs trained on ImageNet are strongly texture biased, whereas human vision is shape biased. For instance, CNNs often classify an image of a "cat" with elephant skin texture as "elephant", showing reliance on local patterns rather than global shape. More recent studies extend this question to VLMs. (Gavrikov et al. (2025)) demonstrate that contrastive multimodal models like CLIP display a higher shape bias than the vision only CNNs, suggesting that pairing images with language directs model's attention to the global object shape. However, VLMs still underperform humans, achieving shape recognition at only 50 - 70% compared to 96% in humans. Critically, they also find that the bias is steerable through language. By modifying prompts the shape recognition shifts from 49% to 72%, underscoring the influence of the text modality on visual biases. Contrary to this, in our experiments, we find that steering does not solve the object bias issue.

## 2.3 OBJECT STATE CHANGE

Early works such as Isola et al. (2015)) introduced the idea of pairing objects with state descriptors (e.g., ripe apple, broken glass) but its coverage was limited and imbalanced. Newman et al. (2024) introduced the ChangeIt-Frames dataset which consists of 25,735 images from instructional videos covering 96 object states. While the dataset covers a wide range of objects, most of these objects are only presented in two states with some of these states being visually very similar. Evaluating nine open source VLMs, they found a consistent drop from object recognition 90-95% to state recognition 60 - 65% in zero-shot setting.

More recent datasets have moved toward video-based settings and finer-grained tasks yet significant gaps remain. Manousaki et al. (2024)) builds on the Ego4D dataset by proposing the OSCA benchmark for anticipating future state changes in egocentric video. While it offers large-scale, real-world data, many of the object categories are represented in only a handful of states. Similarly, Yu et al. (2023) poses state change as a segmentation problem, requiring models to segment objects before and after a transformation. Although it introduces a challenging video segmentation task, the range of objects and states is again limited. Another line of work, Tateno et al. (2025)) tackles multiple object states and their transitions by introducing multi-label annotations for six object categories across 60 state types. This increases state diversity, but at the cost of object coverage, leaving most objects and their transformation unrepresented. Xue et al. (2024)) aims for broader generalization by localizing open-world object state changes in instructional videos. Recent work has also targeted segmentation and manipulation-centric tasks. Tokmakov et al. (2023)) explores how objects undergoing physical transformation challenge standard video object segmentation. Most recently Mandikal et al. (2025)) introduces a new benchmark consolidating prior ideas into a large-scale resource, yet even here the object-state distribution is far from complete.

A common limitation across these datasets is their restricted coverage of object-state combination. Many focus on a small set of objects or a narrow group of states. This creates distributional biases that encourage models to rely on frequent states rather than generalize to unseen transformation. Importantly in the kitchen domain where state changes are both frequent and highly varied diverse object-state transformations annotations remain underrepresented. While datasets contain cooking scenes, annotated coverage of diverse objects and rare cooking transformations is sparse.

In summary, existing benchmarks provide valuable testbeds for evaluating aspects of state change understanding, but none yet achieve broad and balanced coverage across diverse objects and states. This limitation constraints our ability to measure object-state bias in VLMs.

## 3 BBIOS DATASET AND BENCHMARK DESIGN

### 3.1 COLLECTION PROCESS

To ensure a comprehensive evaluation we develop and collect a new dataset allowing us to isolate specific objects/states and curate multiple states per object. As mentioned previously, we focus on objects and states from a single domain, i.e. cooking, so that there is a many-to-many relationship between objects and states. We choose the VidOSC dataset Xue et al. (2024) as a starting point for two reasons: Firstly, frames containing objects and states represent an in-the-wild setting where

| | Fried | Grated | Mashed | Melted | Peeled | Raw | Shredded | Sliced |
|---|---|---|---|---|---|---|---|---|
| Apple | | ✓ | | | ✓ | ✓ | | ✓ |
| Avocado | | | ✓ | | ✓ | ✓ | | ✓ |
| Banana | ✓ | | ✓ | | ✓ | ✓ | | ✓ |
| Carrot | | ✓ | | | ✓ | ✓ | | ✓ |
| Chicken | ✓ | | | | | ✓ | ✓ | ✓ |
| Chocolate | | ✓ | | ✓ | | ✓ | | ✓ |
| Cucumber | | ✓ | | | ✓ | ✓ | | ✓ |
| Egg | ✓ | | | | ✓ | ✓ | | ✓ |
| Eggplant | ✓ | | | | ✓ | ✓ | | ✓ |
| Garlic | ✓ | ✓ | | | ✓ | ✓ | | ✓ |
| Ginger | ✓ | ✓ | | | ✓ | ✓ | | ✓ |
| Lemon | | ✓ | | | ✓ | ✓ | | ✓ |
| Onion | ✓ | ✓ | | | ✓ | ✓ | | ✓ |
| Potato | ✓ | ✓ | ✓ | | ✓ | ✓ | | ✓ |
| Tomato | ✓ | | ✓ | | ✓ | ✓ | | ✓ |
| Zucchini | | ✓ | | | ✓ | ✓ | | ✓ |

**Table 1:** Overview of Objects and States within BBiOS. Each ✓ refers to 10 image samples per object-state pair

objects may not be centered, have a cluttered background, etc. Secondly, videos of state change datasets contain both the initial, 'raw' state of an object and the state after the transition. For example, in a video of an apple being peeled, with the initial state, which we label as 'raw', as well as the other state (e.g. 'peel'), representing the object state change, we can collect two separate states for our benchmark dataset from a single video.

We utilised a multi-stage, semi-automated process for speed and accuracy in creation of the dataset. Firstly, we curated a list of objects and their states from the list of object and states available in the VidOSC, ensuring a many-to-many relationship between objects and states and that for each object the states are visually distinct and can be easily distinguished by a human. Next, we used a Large Language Model (LLM), Llama3.1-70B, to recommend potential frames based on how well they match to the object state(s) within the video. We started by extracting all the frames from the video and then pass each frame to the model, the prompt can be found in Appendix B of the Appendix. The top three frames were then reviewed by a human annotator to select the single frame that best represents the object in the 'raw' state and in the other state. If no suitable frames were found, we discard the video to ensure a high quality overall. In the case where timestamps were not available for the object state changes (i.e. in the VidOSC training set), we adapt the process slightly. The LLM is prompted to instead return the top 10 relevant frames, then CLIP is used to choose the three frames used for manual selection. The combination of LLM and CLIP proved more effective at finding clean frames which showcase the object in the correct state in comparison to solely utilising the LLM across an entire video.

## 3.2 BBiOS Statistics

The collection process resulted in a curated dataset compromising of 16 distinct objects, with each object having between four and six states from a total of eight states resulting in 710 overall images. Importantly, we see BBiOS as a zero-shot evaluation only benchmark for object/state bias. Table 1 shows the combinations of objects and states within BBiOS and Fig. 2 shows examples of images from the dataset. On average, each object contains roughly 45 images, whereas each state contains around 90 images. A full distribution of objects and states within the dataset can be seen in Fig. 7 in the Appendix. We note the non-uniform nature of the dataset due to the differing number of states chosen per object even with the consistent 10 images per object-state combination. This matches similar real-world distributions of classes, e.g. in Damen et al. (2020).

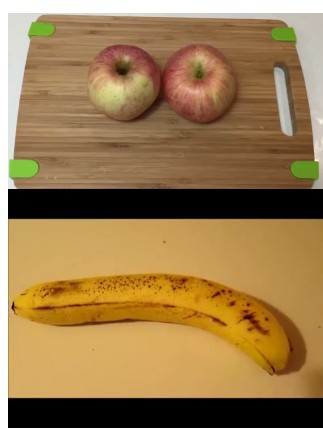
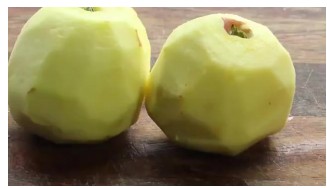
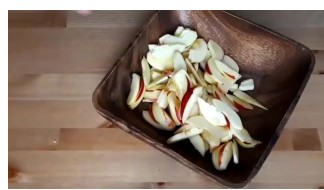
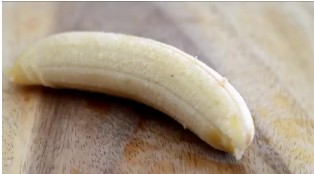
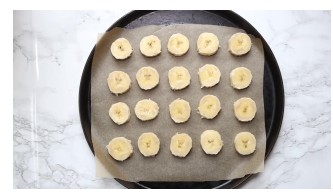

**Figure 2:** Examples of objects and states from the BBiOS dataset (raw, peeled, sliced).

## 3.3 Experimental Design and Metrics

We formulate the experiments as a classification task in which models select the appropriate answer from a set of classes. With dual-encoder (i.e. CLIP) methods, we utilise the zero-shot prediction paradigm from Radford et al. (2021), whereas for LLM-based models we use a closed-answer VQA-style set up as shown in Fig. 1. All LLM generations utilized greedy decoding with a temperature of 0 to maximize determinism and remove stochastic noise. Additionally, we conducted inferences on images as independent instances, avoiding batch processing to prevent potential cross-sample context leakage or parallelization errors. We design six experiments to evaluate how models may be biased towards objects or states. These can be divided into two categories based on whether the model focuses on both the object and state recognition task or is given a forced choice to predict either an object or state. We label these as Multi-Task and Forced-Choice experiments, respectively.

## 3.4 Human Baseline

To establish a baseline and validate the clarity of our dataset, we conducted a human performance study on a representative subset of images. Participants were tasked with identifying both the object and its state, achieving a consistent 87% accuracy across both Object and State Accuracy. This consistent performance demonstrates that object and state recognition are present at an equivalent level of difficulty for human observers. Furthermore, the high baseline confirms that our dataset is not inherently difficult and the the visual features required for classification of both the object and state are distinct and recognizable.

### 3.4.1 Metrics

We evaluate the models using object accuracy and state accuracy, i.e. the accuracy of model at predicting objects or states respectively. To compare the bias of the models, we plot the object accuracy and the state accuracy for a particular model – the distance from the $y = x$ line represents the bias towards either objects or states.

### 3.4.2 Multi-Task Experiments

In this setting, the models are evaluated on a multi-task setup for both object and state recognition, predicting either an object and or a state or a combination of both. More formally, a model, $f$, will predict both an Object $o$, from a set of objects $O$, and a state $s$, from a set of objects $S$ for a given input image $x$ and text $y$, given as: $(o, s) = f(x, y)$. We further sub-divide these experiments based on the level of conditioning the model is given.

**Unconditioned State/Object:** This experiment focuses on evaluating how well the model identifies either the object or the state in isolation, without being influenced by the other. By separating the prediction, we aimed to determine whether the model exhibited any inherent bias towards recognizing objects versus states. Thus, we used one prompt for objects and one for states:

**Objects:** "This is an image of {object}"       **States:** "This is a {state} object"

where {object}/{state} refers to the different options given to the model via substitution.

**Conditioned State/Object:** In this experiment, we inject oracle information of the class not being predicted to see how the models may be influenced – and whether this could be used to debias model predictions. For example, when predicting the object of an image, we give the model the information of the state of the object it is trying to predict. We similarly evaluated two types of prompts:

**Objects:** "This is an image of [GT state] {object}"       **States:** "This is an image of {state} [GT object]"

where [GT object]/[GT state] refers to the GT object/state given to the model.

**Unconditioned Joint Prediction:** Finally, we asked the model to jointly predict the object and state for an image by predicting the tuple $(o, s)$ out of all combinations, i.e. $(o_i, s_j) \in O \times S$. This approach determined whether models were biased towards certain combinations of objects/states and we used the following prompt: "This is an image of {state} {object}"

### 3.4.3 FORCED-CHOICE EXPERIMENTS

In these experiments, we force the models to choose how it classifies an image as an object or a state by giving it all possible options. More specifically, the model $f$ predicts a single class $c$ from the set of all objects and states, i.e. $f(x, y) = (o \vee s) \in \{O \cup S\}$. We can thus determine whether models have a preference for predicting objects or states and can attempt to steer the model via prompting towards predicting a specific class.

**Forced-Choice Control** This experiment acts as the control and highlights the models' preferences on predicting either an object or a state. We use the following prompt: "Which one of these options describes the image correctly."

**Forced-Choice Object Steering:** We next steered the models towards predicting the object within the image under the *forced choice setting* utilising the following prompt: "Which one of these options describes the **primary object** in the image correctly."

**Forced-Choice State Steering:** Finally, we steered the models to predict the state within an image using the following prompt: "Which one of these options describes the **primary state** in the image correctly."

### 3.5 MODEL SELECTION

We conducted our experiments on a diverse set of 23 open source models and 1 commercial model. These models were chosen to represent a wide spectrum of architectures, training paradigms and parameter scale, ranging from small models such as CLIP to large models with up to 90 billion parameters, e.g., Llama3.1. The 23 open source VLMs were selected according to these criteria: **Architecture Diversity:** Covering transformer-based encoders/decoders, dual-encoder and unified multimodal architectures **Parameter Scale:** Spanning small models (<1B parameters), mid-size models (1-20B) and large models (>20B parameters) **Accessibility:** Focusing on models that are publicly available, widely cited and represent different approaches to multimodal learning. This strategy enables comparison not only across models of similar size but also across different design trends, allowing us to isolate the contributions of scale, age, and architecture on the performance. We also included Google Gemini 2.5 Flash-Lite (Google (2025)) to establish a baseline for commercial capabilities. As a representative of state-of-the-art propriety LLMs, Gemini serves as a high-performance benchmark allowing us to contextualize the results of the other selected models against current industry standards.

## 4 RESULTS

### 4.1 MULTI-TASK EXPERIMENTS

The Multi-Task experiments investigate how vision language models handle object and state recognition without explicit guidance. Results of all three sub-experiments can be found in Figure 3.

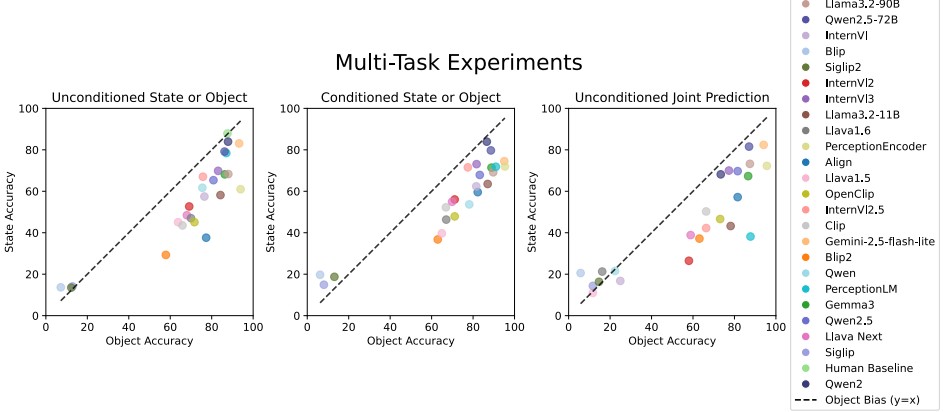

**Figure 3:** Multi-Task Object vs State Accuracy for (left) Unconditioned State/Object, (Middle) Conditioned State/Object, (Right) Unconditioned Joint Prediction.

**Unconditioned State/Object**  When asked to identify objects or states independently, object recognition is seen to outperform state recognition. The majority of models achieve high object accuracies between 60% and 80% while their corresponding state accuracies are substantially lower (40%–70%). Only a small subset of models approached the $y = x$ line with most models falling below it, confirming a strong object bias when only a single piece of information is provided. When compared to the established human baseline, a distinct gap can be observed between human and model performance which suggests that the bias is due to inherit model behaviour rather than ambiguity within the data.

**Conditioned State or Object**  Introducing oracle knowledge for either the object or the state improved overall performance. Most notably, state accuracies increased, sometimes even approaching the object accuracy, suggesting that the knowledge can reduce the ambiguity of the classification. A potential reasoning is that for example when the object is fixed, the model can utilise possible valid states for a given object. For example, if the given ground-truth object is 'chicken', it is (highly) unlikely that the state will be 'melted'. This finding confirms the importance of contextual information and that part of the object bias could be attributed to the way these models resolve ambiguity. However, almost all models still exhibit object bias showcasing that this doesn't solve the problem entirely if the oracle knowledge could indeed be injected.

**Unconditioned Joint Prediction**  When both the object and state are predicted simultaneously, the task becomes significantly harder. The object and state accuracy both drop compared to the conditioned case apart from PerceptionEncoder. Models show difficulty in reasoning about two attributes together and the object bias becomes more evident as the object accuracy consistently outperforms the state accuracy despite the overall reduction. This suggests that when predicting objects and states jointly, models will revert to their stronger representation, i.e., objects, and state predictions become unreliable.

In summary, the Multi-Task experiments show that while conditioning improves the balance between object and state recognition, the bias towards objects remain consistent across these models. State accuracies are consistently underperforming in comparison to object accuracies and this becomes more challenging when varied alongside the object. Whilst injecting oracle information can help overall performance, the object bias across almost all models still exists.

### 4.1.1 TOP-3 INCORRECT PREDICTIONS

In this section, we explore incorrect predictions of models in the *Multi-Task Setting*. We aim to discover whether models' mistakes are biased towards certain classes; whether this is common across different models; and whether this is interconnected across object and state predictions.

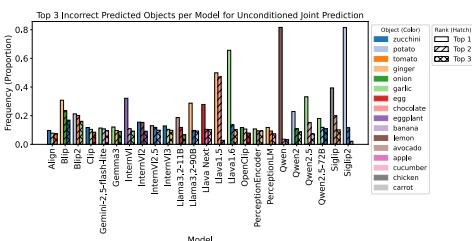

**(a)** Distribution of top 3 incorrect object predictions

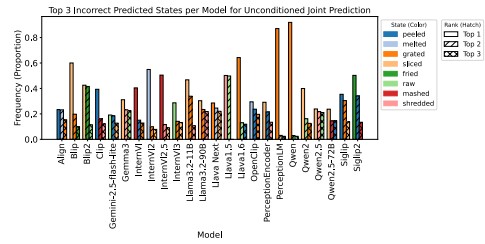

**(b)** Distribution of top 3 incorrect state predictions

**Figure 4:** Top 3 Incorrect predictions of all 24 VLMs across objects (left) and states (right). Results show that models are inconsistent both in their incorrect predictions and the uniformity of these incorrect predictions.

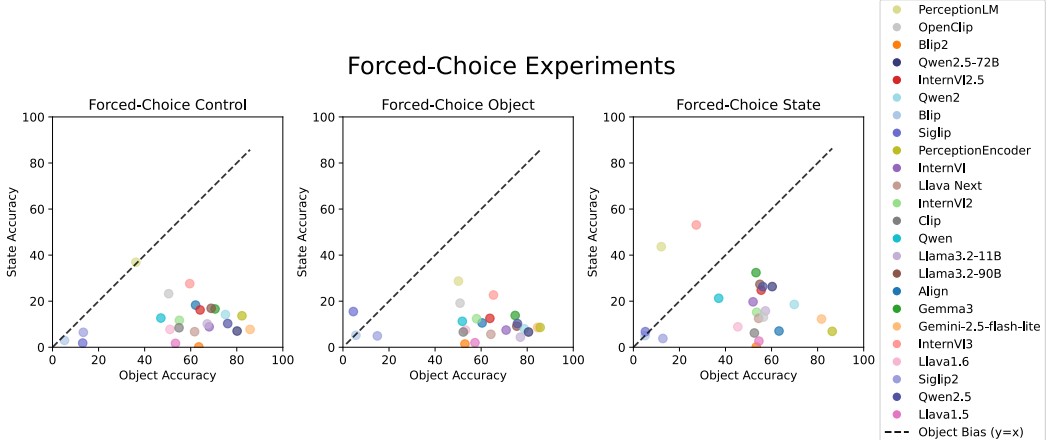

**Figure 5:** Object vs State Accuracy for Forced-Choice Experiments

**Incorrect Object Predictions** Figure 4a presents the top-3 most frequent incorrect object predictions across models. Early models like Align tend to fallback to common objects such as 'potato' and 'tomato'. Clip and Gemma incorrectly predict 'garlic' likely due to 'garlic' being one of the most common cooking ingredients. Qwen defaults to 'lemon' while OpenClip, PerceptionEncoder and PerceptionLM and Gemini 2.5 Flash-Lite are more uniform in their mispredictions.

**Incorrect State Predictions** Figure 4b shows the top-3 most frequent incorrect state predictions across models. We see a similar trend in Align and Blip choosing common states such as 'sliced' and 'fried'. Other models like Gemma favour 'sliced' and 'shredded', reflecting texture preference. Larger, LLM-based models like Qwen mainly default to 'grated' suggesting overfitting. Similar to the incorrect object predictions, Gemini 2.5 Flash-Lite shows a uniform distribution of the mispredictions. Preparation states dominate, indicating data bias, while states like 'melted' are underrepresented suggesting reasoning gaps.

Overall, the results reveal that methods are not consistent in their mis-classifications, i.e., there is not one or two objects/states that are consistently predicted across all models. Additionally, whilst some models tend to over-predict certain classes, others are more uniform. These trends can be seen across older/newer models and dual encoder/LLM-based models suggesting that these biases are still active issues to solve. These highlight poor training data diversity and weak feature extraction across models. Improving these aspects could reduce errors, particularly in the joint prediction task where compounding biases can amplify the misclassifications.

## 4.2 FORCED-CHOICE EXPERIMENTS

The Forced-Choice experiments in Figure 5 extend the analysis by introducing model preference for object and states in addition to performance by forcing the models to choose only an object or state class for each image.

**Forced-Choice Control** We see that the object accuracy results are largely similar to the multi-task experiments, yet the state accuracy suffers a huge drop due to the forced choice. Only PerceptionLM is able to achieve similar object and state accuracies, yet its object accuracy falls behind many of the other models by over $40\%$. Additionally, we find that models overwhelmingly default to predicting objects over states, on average models predict objects for $75\%$ of images.

**Forced-Choice Object** When explicitly directed to prioritize the object, models maintained high performance in object classification and slightly increase their preferences for predicting objects to $78\%$. In fact, the steering prompt reinforces the models' behaviour, pushing them to focus more on objects and in many cases the state accuracy drops even if the object accuracy does not improve by much. Overall, the object bias largely either remains the same, or increases dramatically, in the case of PerceptionLM, Align, and InternVL3.

**Forced-Choice State** When models are steered towards providing a state description for the image, state accuracies tend to increase slightly. However, the state accuracy is again much lower than the object accuracies across all but two models: InternVL3 and PerceptionLM. Interestingly, both of these models showcase strong steering capabilities, yet still predict objects with the state steering prompt and showcase a large drop in object performance when doing so. Otherwise, the remaining models have a preference towards predicting the object $68\%$ of the time – showcasing that the models are still heavily object biased and not directly answering the question of the primary state of the object in the image. These findings suggest that steering can partially improve the gap, but that the root of the bias lies in the models' underlying representation of states and the training data utilised.

### 4.2.1 CHANGE IN PERCENTAGE OF STATE PREDICTIONS

We showcase in Figure 6 the percentage change in number of state predictions within the Forced-Choice experiments, comparing the object and state steering to the control experiment. The results demonstrate substantial variability in model behaviour when steering prompts are used to emphasize either object or state with some patterns emerging from the data.

The biggest increase was SigLip, which showed more than 60% increase in state predictions compared to the control, suggesting that object and state representations are more inter-connected in its representation space. Otherwise, most models saw an increase in state predictions when steered, including InternVL, Qwen2.5-72B and PerceptionLM. Across most models, state-focused steering produced the expected positive increase in state predictions, ranging from approximately 8% to 35%. Models such as Qwen2.5-72B, InternVL, and PerceptionLM showed robust positive increase indicating successful steering toward state-based predictions, for the latter two models, this matches their ability to become state biased models. Object-focused steering generally produced smaller magnitude changes compared to state focused steering, with most models showcasing a slight increase in predicting objects. However, as noted above, whilst the preference changed, the biases did not across 21 out of the 24 models. This furthers our finding that object bias is inherently a representation and training data issue.

### 4.2.2 ABLATION STUDY: PROMPT TUNING

To distinguish between limitations inherent to the model representations and potential artifacts of prompt engineering, we conducted a series of ablations targeting the semantic framing of our prompts. Specifically, we evaluated the robustness of our approach by substituting key terminology with semantically related alternatives. We focused on the two primary components of our baseline prompts:

**State Terminology:** We replaced the term "State" with "State Change", "Transition", "Condition" and "Transformation".
**Object Terminology:** We replaced the term "Object" with "Ingredient".

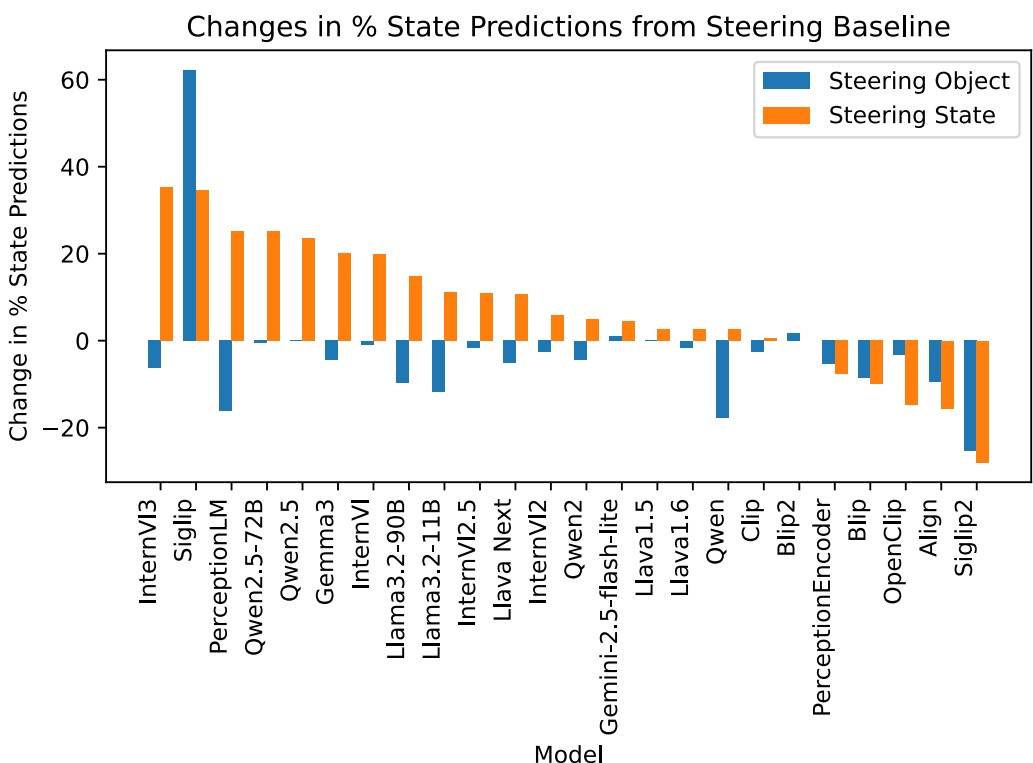

**Figure 6:** Percentage change in number of state predictions for Forced-Choice experiments

The substitution of object with ingredient yielded negligible changes in performance. Altering the terminology for state resulted in noticeable performance fluctuation. While certain synonyms improved steering capabilities for specific architectures, they degraded performance in others. Notably, no single prompt variation consistently outperformed the original baseline across all the evaluated architectures. The lack of a universally superior prompt suggests that the observed performance is likely intrinsic to the models' internal representation rather than a result of suboptimal prompt tuning. Detailed results of these ablations are provided in the Supplementary Material C.

## 5 CONCLUSION

In this work, we introduced the Benchmark for Biases in Objects and States (BBiOS) and used it to systematically examine how Vision-Language Models attempt to balance object and state recognition. Across the multi-task and forced-choice experiments, our analyses reveal a clear and consistent object-state bias: models consistently recognize object categories but are noticeably less reliable with state recognition, even when explicitly steered through prompting or injected with oracle knowledge. While conditioning and steering strategies can nudge model behaviour, their effects are limited and inconsistent, reinforcing the idea that object bias is inherent to the models overall.

These findings suggest that the challenge lies on deeper aspects of model training and architecture. Addressing object-state bias may require novel multimodal objectives, richer datasets that emphasize state variability, or architectural changes that disentangle reasoning between objects and their states. More broadly, our results highlight a critical gap in multimodal reasoning: the ability to integrate object identity with dynamic states in a robust and generalizable manner. We hope that (BBiOS) serves as a foundation for future work aimed at designing models that holistically understand objects and their states.

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

APPENDIX

## A MODELS

We list below all the models used during our experiments. Experiments were carried out using two NVIDIA GH200.

| Model Name | Checkpoint | Transformation |
|---|---|---|
| Clip Radford et al. (2021) | `ViT-B/32` | default transformations |
| OpenClip Cherti et al. (2023) | `ViT-B-32` | default transformations |
| ALIGN Jia et al. (2021) | `kakaobrain/align-base` | default transformations |
| BLIP Li et al. (2022) | `Salesforce/blip-image-captioning-base` | default transformations |
| BLIP-2 Li et al. (2023) | `Salesforce/blip2-itm-vit-g` | default transformations |
| SigLIP Zhai et al. (2023) | `google/siglip-base-patch16-224` | default transformations |
| SigLIP2 Tschannen et al. (2025) | `google/siglip2-base-patch16-224` | default transformations |
| Perception Encoder Bolya et al. (2025) | `PE-Core-L14-336` | default transformations |
| Qwen – 10B Bai et al. (2023) | `Qwen/Qwen-VL-Chat` | default transformations |
| Qwen2 – 8B Wang et al. (2024a) | `Qwen/Qwen2-VL-7B-Instruct` | default transformations |
| Qwen2.5 – 8B Team (2025) | `Qwen/Qwen2.5-VL-7B-Instruct` | default transformations |
| Qwen2.5 - 72B Team (2025) | `Qwen/Qwen2.5-VL-72B-Instruct` | default transformations |
| InternVL – 19B Chen et al. (2024b) | `OpenGVLab/InternVL-Chat-V1-1` | default transformations |
| InternVL2 – 8B OpenGVLab (2024) | `OpenGVLab/InternVL2-8B` | default transformations |
| InternVL2.5 – 8B Chen et al. (2024a) | `OpenGVLab/InternVL2_5-8B` | default transformations |
| InternVL3 – 38B Zhu et al. (2025) | `OpenGVLab/InternVL3-38B-Instruct` | default transformations |
| LLaVA NeXT – 8B Li et al. (2024a) | `llava-hf/llama3-llava-next-8b-hf` | default transformations |
| LLaVA1.5 – 7B Liu et al. (2024a) | `llava-hf/llava-1.5-7b-hf` | default transformations |
| LLaVA1.6 – 7B Liu et al. (2024b) | `llava-hf/llava-v1.6-vicuna-7b-hf` | default transformations |
| PerceptionLM – 8B Cho et al. (2025) | `facebook/Perception-LM-8B` | default transformations |
| Gemma3 – 12B Team et al. (2025) | `google/gemma-3-12b-it` | default transformations |
| Llama3.2 - 11B Meta (2024) | `meta-llama/Llama-3.2-11B-Vision-Instruct` | default transformations |
| Llama3.2 - 90B Meta (2024) | `meta-llama/Llama-3.2-90B-Vision-Instruct` | default transformations |
| Google Gemini 2.5 Flash-Lite Google (2025) | `gemini-2.5-flash-lite` | default transformations |

**Table 2:** List of Models used during experiments

## B DATASET

We will release the dataset images and the accompanying benchmark code for evaluation once the reviewing process has concluded.

For the frame selection using an LLM, we used the below prompt to score each image on a scale from 1-10. We tested different prompts and were able to empirically validate that this prompt provides the least number of false positives and ensuring a high quality of images provided.

```
You are an expert image analyst. Your task is to determine
how well this image represents the EXACT object and state:
``{object} {state}''

CRITICAL INSTRUCTIONS:
1. BE EXTREMELY PRECISE about the state - ``{state}''
is the EXACT state we need
2. If the image shows {object} in a DIFFERENT state, give a
LOW score (0-3)
3. If the image has NO {object} at all, give score 0
4. Only give HIGH scores (7-10) if the state and object matches
EXACTLY
5. Partial matches or similar states should get MEDIUM scores (3-6)

Analyze this image and provide response in this EXACT JSON format:
{{
    ``confidence_score'': <number from 0-10>,
    ``object_state_observed'': ``<describe the actual state of
    {object} if present>''
}}
```

```
Remember: Be strict about the EXACT object and state match.
Only high scores for exact matches!
Only respond with valid JSON, no other text.
```

A more detailed plot of the distribution of the dataset for each object and state can be seen in Fig. 7. As previously mentioned we note the non-uniform distribution of both the objects and states.

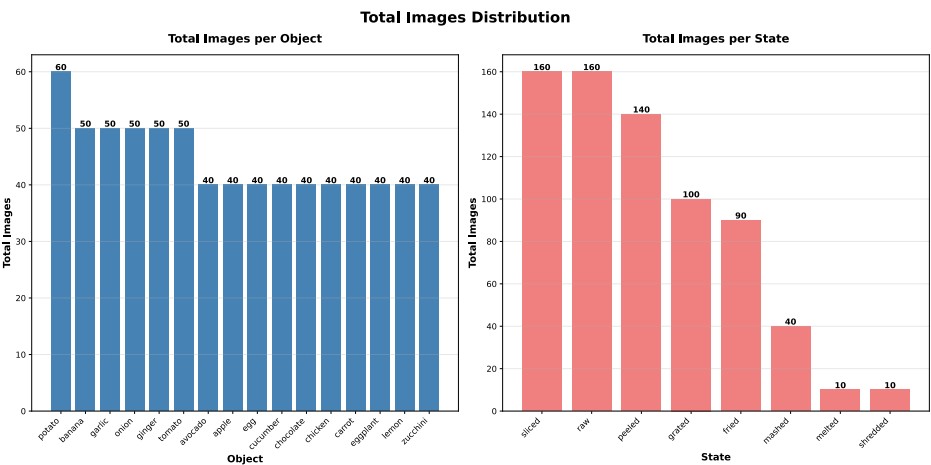

**Figure 7:** Distribution of objects and states

## C  ABLATIONS

**Forced-Choice Ingredient**  The Forced-Choice Ingredient plot mirrors the Control setting, with the vast majority of models remaining clustered in the lower-right quadrant. This implies that the concept of an ingredient is likely semantically coupled with the object's identity rather than its state within the models' latent space. Consequently this specific keyword fails to steer the models' attention toward state-differentiation features, resulting in continued high performance on object recognition.

**Forced-Choice Transformation**  The introduction of the "Transformation" keyword causes a significant shift in the models' performance with several models moving closer to or even crossing the object bias line. This suggests that the term "Transformation" acts as a successful cue, encouraging the model to focus on features related to change. Despite this positive trend, the steering in not universally effective; some models' state accuracy performance decreases.

**Forced-Choice Transition**  When the prompt is altered to use the word "Transition", there is a clear shift toward better state recognition, with many models clustering near the diagonal. Despite this trend, a distinct subset of models performs worse. A cluster of models remains pinned to the bottom-right (high object/low state accuracy), indicating that these specific architectures are unaffected by the replacement of "State" with "Transformation". Furthermore, some models that had moderate object accuracy in the Control setting see a drop in object accuracy without a significant gain in state accuracy, suggesting that the steering prompt likely confused the model rather that effectively redirect the models attention.

**Forced-Choice Condition**  In the Forced-Choice Condition plot, results revert toward the baseline, implying that "Condition" is a weak steering term.

**Forced-Choice State Change**  The Forced-Choice State Change plot demonstrates that while this phrase is generally an effective steering prompt, it induced high variance and negatively impact specific models. While many models rise to the top-left, a noticeable group remains trapped in the bottom-right corner, with some models performing even worse on state accuracy

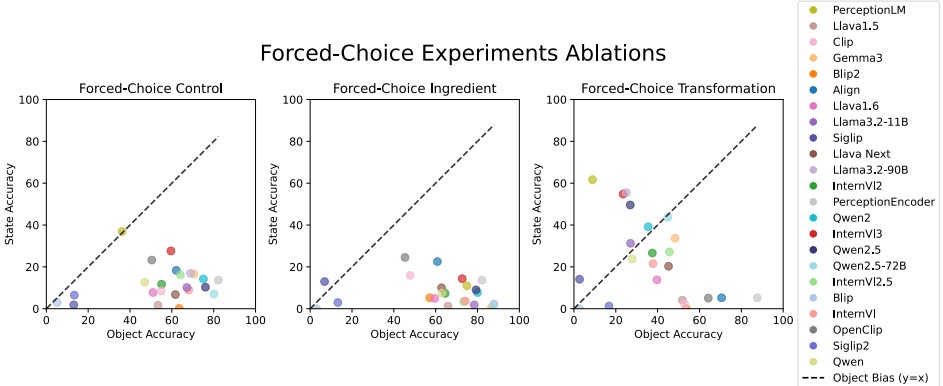

**Figure 8:** Multi-Task Object vs State Accuracy for (left) Forced-Choice Control, (Middle) Forced-Choice Ingredient, (Right) Forced-Choice Transformation.

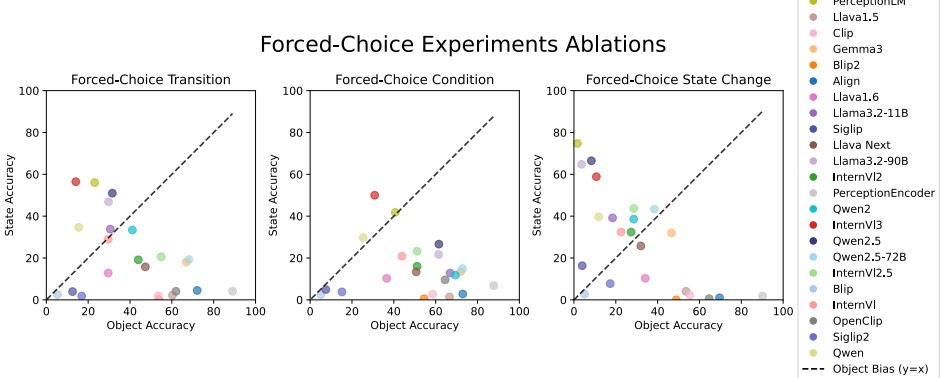

**Figure 9:** Multi-Task Object vs State Accuracy for (left) Forced-Choice Transition, (Middle) Forced-Choice Condition, (Right) Forced-Choice State Change.

# D PROMPTS

This section provides examples for all the prompts used during our study.

## D.1 UNCONDITIONED STATE

```
Which one of these options describes the image correctly.

1. sliced
2. melted
3. shredded
4. grated
5. fried
6. raw
7. peeled
8. mashed

Answer ONLY with the exact text from one of the above choices.
Your response MUST be one of these options. DO NOT make up words that are not in the
```

## D.2 UNCONDITIONED OBJECT

```
Which one of these options describes the image correctly.
```

```
1.  This is an image of a onion.
2.  This is an image of a apple.
3.  This is an image of a cucumber.
4.  This is an image of a egg.
5.  This is an image of a lemon.
6.  This is an image of a carrot.
7.  This is an image of a avocado.
8.  This is an image of a potato.
9.  This is an image of a eggplant.
10. This is an image of a tomato.
11. This is an image of a garlic.
12. This is an image of a zucchini.
13. This is an image of a ginger.
14. This is an image of a chocolate.
15. This is an image of a banana.
16. This is an image of a chicken.

Answer ONLY with the exact text from one of the above choices.
Your response MUST be one of these options.
DO NOT make up words that are not in the answers"
```

### D.3 CONDITIONED OBJECT

```
Which one of these options describes the image correctly.

1. This is an image of fried {object}.
2. This is an image of grated {object}.
3. This is an image of shredded {object}.
4. This is an image of raw {object}.
5. This is an image of sliced {object}.
6. This is an image of melted {object}.
7. This is an image of mashed {object}.
8. This is an image of peeled {object}.

Answer ONLY with the exact text from one of the above choices.
Your response MUST be one of these options.
DO NOT make up words that are not in the answers
```

### D.4 CONDITIONED STATE

```
Which one of these options describes the image correctly.

1.  This is an image of {state} chocolate.
2.  This is an image of {state} eggplant.
3.  This is an image of {state} cucumber.
4.  This is an image of {state} garlic.
5.  This is an image of {state} tomato.
6.  This is an image of {state} lemon.
7.  This is an image of {state} avocado.
8.  This is an image of {state} ginger.
9.  This is an image of {state} banana.
10. This is an image of {state} potato.
11. This is an image of {state} egg.
12. This is an image of {state} carrot.
13. This is an image of {state} apple.
14. This is an image of {state} zucchini.
15. This is an image of {state} chicken.
16. This is an image of {state} onion.
```

```
Answer ONLY with the exact text from one of the above choices.
Your response MUST be one of these options.
DO NOT make up words that are not in the answers
```

### D.5   UNCONDITIONED JOINT PREDICTION

```
Which one of these options describes the image correctly.

1.  This is an image of {state} {object}.
.
.
.
128. This is an image of {state} {object}.

Answer ONLY with the exact text from one of the above choices.
Your response MUST be one of these options.
DO NOT make up words that are not in the answers
```

### D.6   FORCED-CHOICE CONTROL

```
Which one of these options describes the image correctly.

1.  egg
2.  garlic
3.  potato
4.  lemon
5.  avocado
6.  tomato
7.  zucchini
8.  chicken
9.  carrot
10. chocolate
11. cucumber
12. ginger
13. eggplant
14. apple
15. onion
16. banana
17. fried
18. raw
19. peeled
20. sliced
21. grated
22. shredded
23. melted
24. mashed

Answer ONLY with the exact text from one of the above choices.
Your response MUST be one of these options.
DO NOT make up words that are not in the answers.
```

### D.7   FORCED-CHOICE *Keyword*

```
Which one of these options describes the primary {keyword} in the image correctly.
1.  egg
2.  garlic
3.  potato
```

```
4.  lemon
5.  avocado
6.  tomato
7.  zucchini
8.  chicken
9.  carrot
10. chocolate
11. cucumber
12. ginger
13. eggplant
14. apple
15. onion
16. banana
17. fried
18. raw
19. peeled
20. sliced
21. grated
22. shredded
23. melted
24. mashed

Answer ONLY with the exact text from one of the above choices.
Your response MUST be one of these options.
DO NOT make up words that are not in the answers.
```

