# OpenReview forum: "SEEING THROUGH LANGUAGE: HOW TEXT REVEALS OBJECT AND STATE BIAS IN VLMS"
_ICLR.cc/2026/Conference — Submitted to ICLR 2026_

### Official Review · Reviewer_x27o · 2025-10-27

**Soundness:** 2
**Presentation:** 2
**Contribution:** 1
**Rating:** 0
**Confidence:** 2

**Summary:**

The paper presents an empirical study of VLM object and state recognition capabilities. To this extend, the authors introduce a novel benchmark (BBiOS) for the measurement of the introduced "object-state bias".  The benchmark dataset contains images of different vegetables and fruits at various states of processing (like raw, peeled, sliced, fried ...). The benchmark task is then to predict the object categories and their states. The experimental on this dataset evaluation shows that current (open) VLM models are better in predicting object categories than their state.

**Strengths:**

The paper is well written and easy to follow. The experiments use a large number of open VLMs

**Weaknesses:**

The reviewer strongly disagrees with the way the authors use the term "bias". While the investigation of all sorts of biases is an important ongoing topic, its inflationary usage as buzzword does not provide real insights. The conducted experiments show that VLMs are better in predicting categories than states (for a very specific dataset). It totally unclear how this represents a systematic model bias. While the formal definition of biases might be a bit weak, it always includes a clear relationship (dependency) between variables. The famous shape-texture bias for example, shows a trade-off between texture and shape information used by CNNs. A racial bias leads to model preferences towards people of a certain skin color (at expense of others). In the presented case, it remains unclear how the ability of models to predict object categories would limit the state predictions.

Taking away the bias claim, the paper simply shows that VLMs are better in predicting fruit types than fruit states - this is not especially surprising since the state prediction task is much more complicated (given the variance of states in images).

**Questions:**

* What is your definition of BIAS ?
* the paper only shows results for open VLM models. How do state of the art commercial models like GPT-5 behave?

---

> ### Author Response · Authors · 2025-11-25
> **Response**
>
> # Weakness 1 and Question 1
> We respectfully disagree that our usage of the term "bias" is inflationary, rather it describes a systematic hierarchical feature dependency similar to the well-established shape-texture bias in CNNs. We clearly establish a relation between an object and its state and how the interplay of these two attributes affects model performance. We would also like to note that other reviewers did not believe this to be an issue. Reviewer qmb8 mentions that "Bias investigation is systematically conducted" and reviewer UCUV mentions that we show a "Consistent empirical evidence showing strong object bias". The reviewer suggests that state prediction is simply "harder" due to variance yet our human baseline shows similar performance across both object and state showcasing an intrinsic equal difficulty in humans, yet our comprehensive experiments do not show the same for models. This demonstrates that the performance gap is not a function of task difficulty but of semantic relevance.
>
> # Question 2
> We have now extended our evaluation to include Gemini 2.5 Flash Lite. Our results indicate that this commercial model exhibits the same performance trend observed in the top-performing open models in our paper: The results are summarized in the table below
>
> | Experiment | State Accuracy % | Object Accuracy% |
> | ------------- | ------------- | ------------- |
> | Unconditioned State/Object      | 83 | 93 |
> | Conditioned State or Object      | 75 | 95 |
> | Unconditioned Joint Prediction  | 82 |  94 |
> |Forced-Choice Control                | 7 |  86 |
> | Forced-Choice Object                |  9 |  85 |
> | Forced-Choice State                  | 12 | 82 |
>
> However, consistent with our other findings, the object accuracy remains higher than state accuracy. We will update the final version of the paper to include these results.

---

### Official Review · Reviewer_qmb8 · 2025-10-31

**Soundness:** 3
**Presentation:** 3
**Contribution:** 2
**Rating:** 4
**Confidence:** 5

**Summary:**

This submission studies the object-state bias in vision-language models (both dual-encoder and autoregressive). The results are based on the collected and annotated BBiOS dataset, containing 16 kitchen objects with 4-6 states each (including one raw state) from a total of 8 states. A study on 23 different models then evaluates the object-state bias in multi-task and forced-choice settings and reveals that 1) models struggle at state recognition as compared to objects; 2) are biased toward objects; and 3) are mostly steerable in the state direction and sometimes not at all.

**Strengths:**

- Introduces new “Bias in Object State (BBiOS)” dataset based on kitchen ingredients (derived from VidOSC) consisting of 16 objects and 4-6 states each (including one raw state) from a total of 8 states to evaluate object and state bias claiming a balanced coverage across object-state pairs
- Bias investigation is systematically conducted in multi-task and forced choice settings
- Evaluation of 23 diverse VLMs
- Prompt-based steering is evaluated and shown to less effective than shown in previous studies on other cues

**Weaknesses:**

- Previous bias studies have been based around the methodology that all cues are easily individually recognizable. Yet, models seem to struggle to detect state (as also shown by Newman et al. (2024))–this may introduce a confounder, i.e., models may not be biased–they simply do not recognize the correct state (and then potentially hallucinate the state or fall back to object-only responses).
- Prompts for “steering” are not tuned; in general no prompt seems to be tuned which may bias the study. For example, the terms "state" and "object" may be simple poorly aligned in VL spaces. It would be good to ablate a few other prompts to show that the findings are not limited by the choice of prompt.
- The prompt template for LLMs is not properly described (Sec. 3.3.2 is lacking details). A few examples would be helpful.
- Also, it is not described how LLM responses are sampled. Using non-greedy sampling may have introduced significant error and mandates a statistical analysis of error between sampled responses in parallel.
- If LLMs are poor at state recognition (Newman, 2024) then they should also introduce an error in the data curation pipeline (L192ff)
- Non-uniform distribution of object/state and  the paper is missing a heatmap two show the joint frequency of object-state pairs (or alternatively numbers/ratios in Table 1)
- The models are poorly documented. Please precisely name the checkpoint (e.g, there are multiple SigLIP variants, Qwen – 10B is probably Qwen-Image-10B etc.), weight source, and inference resolution (transformations)

Minor:
- Figures are not vector graphics and blurry
- LLaVa -> LLaVA (L99)
- Unnecessary parentheses around citep in intro and elsewhere
- Inconsistent reference capitalization (Fig/fig, e.g, L213; sometime abbreviated or not)

**Questions:**

- Does in-context learning or chain-of-thought change any of the results?

---

> ### Author Response · Authors · 2025-11-25
> **Response Part 1**
>
> # Weakness 1
> Our methodology explicitly controls for this confounder through the experimental design. Our experiments decouple object and state recognition (see section 3.3.2). By providing the ground truth object (or state) in the prompt, we effectively remove the recognition burden for that variable. Our results show that while providing ground truth improves state accuracy it does not solve the problem. As for the case of a default fallback pattern, our analysis of the top-3 incorrect object and state predictions in Fig. 3 shows no statistical evidence that suggests that the models systematically fall back to a specific object or state. The distribution of errors is non-uniform indicating that the model does not fallback to a specific object or state.
>
> # Weakness 2
> To address this, we conducted additional ablation studies replacing the term "state" with semantically related alternatives, including "state change", "transition", "condition" and "transformation" and the term "object" with "ingredient". For the case of replacing object with ingredient there was very little change in the performance, however, we observed performance fluctuations for changing the term state. Even though certain prompts improved steering for specific models these same prompts often degraded the performance of others. Consequently, we found no single prompt that consistently outperformed the original baseline across all architectures. This suggests that the limitations we observed are likely intrinsic to the models' representation rather than an artifact of poor prompt tuning. We will add these ablations to the Supplementary Material in the final version.
>
> # Weakness 3
> Below is an example of the prompt for the "Unconditioned State" experiment. For other experiments in section 3.3.2, the choices change depending on the experiment.
> ```
> Which one of these options describes the image correctly.
>     1. sliced
>     2. melted
>     3. shredded
>     4. grated
>     5. fried
>     6. raw
>     7. peeled
>     8. mashed
> Answer ONLY with the exact text from one of the above choices. Your response MUST be one of these options. DO NOT make up words that are not in the answers"
> ```
> # Weakness 4
> We employed greedy decoding (temperature = 0) for all LLM responses. This was done to ensure the models output is deterministic and eliminate variance associated with stochastic sampling methods. Additionally, all images were processed as independent instances to prevent any potential issues due to batch processing or parallelization.
>
> # Weakness 5
> The LLM was employed solely as a preliminary filter to identify candidate frames from the video. Crucially, every frame selected for the final dataset underwent manual inspection and verification by the authors. Therefore, any potential state recognition errors introduced during the automated filtering stage were identified and excluded during the human verification phase, ensuring the integrity of the final data.
>
> # Weakness 6
> To clarify, in our dataset every object-state pair contains exactly 10 images. We apologize for the confusion in Table 1, each tick refers to 10 images. Detailed counts for the breakdown of each object and state is in Fig. 7 in Appendix B. To prevent future confusion we will update Table 1 to explicitly state that each tick represents 10 images.
>
> # Weakness 7
> Thank you for your valuable feedback. We appreciate you pointing out the missing details of our model documentation. In the final version, we will include a detailed table specifying the exact checkpoints names and other relevant information. We also thank you for catching the formatting issues. All figures will be replaced with higher quality version and all formatting issues will be reflected in the final version.

---

> ### Author Response · Authors · 2025-11-25
> **Response Part 2**
>
> # Question 1
> We conducted a preliminary study using our biggest model Llama3.2-90B. We implemented In-Context Learning by providing few-shot examples, varying context to include examples while providing the object and state. Our results show very inconsistent behavior and fluctuations, where the model ignores the provided instruction to provide either the object or the state and returns both the object and state. This led to a change of State Accuracy 17%, Object Accuracy 69 % to State Accuracy 36% Object Accuracy 7\%, which provided a slightly better but still low performance for the extra computational cost, compared to using prompting, State Accuracy 27% Object Accuracy 55%. Given the minimal and unstable gains observed in this pilot test, and the significant computational cost of scaling this across all 23 models, we did not extend this analysis to the full suite of models. As for Chain-of-Thought, given the high computational cost of reasoning we tested our pilot on the smaller version Llama3.2-11B. Our results show that the model performance drops as a result of the reasoning. This was observed in a change of State Accuracy 10%, Object Accuracy 67% to State accuracy 1% Object accuracy 44%. As we inspected the reasoning logic we found that it was inconsistent with regards to focusing on the object or the state which leads to the overall drop in performance. Similar to the In-context Learning pilot, given the drop in performance and the significant computational cost of scaling this across all 23 models, we did not extend this analysis to the full suite of models.

---

> > ### Comment · Reviewer_qmb8 · 2025-11-27
> >
> > Thanks for the rebuttal. First, I'd like to remind the authors that they can and should update the PDF during the discussion phase. I cannot judge what I do not see.
> >
> > W1: The Unconditioned State/Object Accuracy in Fig. 3 shows that objects are easier to predict than states. This is not a bias, this is just a matter of perception.
> >
> > W2: I appreciate the experiment, but I cannot judge it without seeing the numbers. Further, synonyms are only a part of the prompt. The prompt structure itself, the wording, and possible constraints all influence biases and accuracy.
> >
> > W3: Thanks. Can you please provide these for all sub-experiments? It would be great to include these in the appendix.
> >
> > W4: Good. Please add these details to the paper.
> >
> > W5: Okay.
> >
> > W6: Please add this clarification to the paper.
> >
> > W7: I am a glad that I could help, but again: Please add the changes now.
> >
> > Q1: Thanks for the experiments. To clarify: I did not expect a full eval of all models. Could you clarify how exactly you implemented ICL? Also, CoT is typically only effective in large models, but don't worry about adding this experiment.

---

> ### Author Response · Authors · 2025-12-03
> **Response Part 3**
>
> Thank you for your feedback. We will upload an updated version of the paper with all the requested changes, new experiments, improved model summary and the prompts used in the different experiments. The changes are highlighted in red for easier visibility. Regarding the other questions you had:
>
> # Weakness 1
> We define “bias” here as the model relying on the object’s identity to guess the state rather than looking at the visual evidence of the state itself. If the issue were solely that states are “hard to see”, we would expect the model to struggle with specific states consistently. However as shown in Fig. 3 there are no consistent states that the model gets wrong. Given that the same state is seen across different objects, yet the model’s performance drastically changes per model, the error likely stems from each model’s bias toward the object-state relationship.
>
> # Weakness2
> We added new figures to display these results. Regarding the prompt structure, our decision to use a simplified template was deliberate. Since the task is straightforward image classification we wanted to minimize potential noise that could be introduced by elaborate prompt structures. Given the high variance that prompt phrasing introduces and the wide range of models we tested, a complex prompt could have become a confounding variable. By keeping the structure constant and simple, we were able to isolate the ‘keyword’ variable. This allowed us to measure the specific marginal impact of synonyms on performance without the results being skewed by how different models interpret complex prompts.
> # Question 1
> For ICL we used few-shot multimodal in-context learning where we constructed a set of five examples where each context example consisted of the an Image paired with the object and the state of this image. We randomly sampled from our dataset with the requirement that we do not use samples that have the exact same object and state to prevent the model from exploiting semantic similarity from the context.

---

### Official Review · Reviewer_UCUV · 2025-10-31

**Soundness:** 2
**Presentation:** 3
**Contribution:** 2
**Rating:** 4
**Confidence:** 3

**Summary:**

This paper investigates object–state bias in Vision–Language Models (VLMs), where models tend to recognize object identity (e.g., "apple") more accurately than object state (e.g., "sliced", "peeled"). The authors introduce the Benchmark for Biases in Objects and States (BBiOS), a curated image dataset of 16 kitchen objects across eight states, collected semi-automatically from VidOSC using LLaMA-3.1 and CLIP for frame selection. The study designs two experimental paradigms—Multi-Task (object/state prediction with or without conditioning) and Forced-Choice (object vs. state classification)—and evaluates 23 diverse VLMs, including CLIP, BLIP, Qwen2-VL, InternVL, and LLaVA families. Results show consistent bias: models achieve high accuracy for object identity but struggle with state recognition, even under oracle conditioning or steering prompts. The paper concludes that object bias is structural and arises from training data and representation design rather than prompting limitations .

**Strengths:**

1. A well-constructed dataset focusing on controlled object–state variations in realistic visual contexts.
2. Comprehensive evaluation across 23 models spanning diverse architectures and parameter scales.
3. Consistent empirical evidence showing strong object bias and limited steerability through text prompts or oracle information.
4. Thoughtful discussion linking dataset composition, linguistic priors, and multimodal representations.

**Weaknesses:**

1. The contribution is primarily diagnostic. The paper identifies object bias but does not analyze its causal origin in vision encoders, textual priors, or training objectives.
2. The dataset size is small for large-scale model evaluation, making the generalization of conclusions uncertain.
3. The evaluation lacks human baselines or psychometric reliability checks to contextualize bias severity.
4. The analysis does not disentangle dataset bias (frequency imbalance) from representation bias (model internal weighting).
5. The paper reuses known conceptual framing (object vs. state) without deeper theoretical grounding—similar trends have been observed in ChangeIt-Frames and recent multimodal perception studies [1–3].

[1] Newman et al., “Do Pre-Trained Vision-Language Models Encode Object States?” arXiv 2024. \
[2] Y. Fu et al., “BLINK: Multimodal Large Language Models Can See but Not Perceive,” ECCV 2024. \
[3] Kawaharazuka et al., “Continuous Object State Recognition for Cooking Robots,” IEEE RAL 2024.

**Questions:**

1. How does the bias magnitude correlate with visual encoder scale or architecture type (ViT-L vs. Swin)?
2. Do object–state biases persist if the textual prompt explicitly disambiguates the state (e.g., “a peeled apple on a plate”)?
3. Could linear probing or concept subspace analysis (e.g., CAV [4]) help localize the state-sensitive directions in the representation?
4. How balanced is the BBiOS dataset in terms of background and lighting? Could visual confounds explain part of the state gap?
5. Is there evidence that models trained on state-rich datasets (e.g., Something-Something V2) show reduced object dominance?

[4] B. Kim et al., “Interpretability Beyond Feature Attribution: Quantitative Testing with Concept Activation Vectors (TCAV),” ICML 2018.

---

> ### Author Response · Authors · 2025-11-25
> **Response Part 1**
>
> # Weakness 1
> While we acknowledge that this work focuses on identifying and quantifying object bias rather than dissecting its specific causal mechanism, we believe this diagnostic contribution is critical precisely because of its scale. By testing 23 models across varied architectures and parameter counts, we demonstrate that this bias is not an artifact of a specific encoder, training objective or model size, but rather a consistent behavior in current vision-language models. Our comprehensive analysis rules out the hypothesis that this is bias is specific to architectural choices. Establishing this is a necessary prerequisite for future analysis, we have effectively defined the 'what' and 'where' so that future work can target the 'why'.
>
> # Weakness 2
> The scope of the dataset was an intentional design choice rather than a limitation. Constraining the domain allowed for more precise annotation and more focus on the fine-grained interactions between object and state. The primary motivation for selecting the kitchen domain is that it uniquely features objects that exist in multiple, visually distinct states (e.g. a whole onion vs diced onion). Since objects in the kitchen undergo drastic visual transformation while maintaining their semantic identity this allows us to rigorously test if VLMs can generalize across visually distinct states of the same object as well as the models understanding of an object and its different states.
>
> # Weakness 3
> We have now conducted a new human validation experiment to establish these baselines. We asked human participants to identify the object and state of a subset of images in our dataset. The results demonstrate a high human baseline, with object accuracy at 87% and a state accuracy of 87%, which is a much higher state accuracy than almost all of the models tested. This confirms that the dataset's ground truth is unambiguous to human observers and that humans find each task to have a similar difficulty. These new results contextualize the bias severity: the gap between the human baseline and the models' performance highlights that the bias is algorithmic and not a result of data ambiguity.
>
> # Weakness 4
> Our evaluation methodology isolates model performance at the instance level, effectively decoupling it from global dataset frequency statistics. By testing images independently, we observe that the model's failures do not align with dataset dominance. Specifically, if dataset bias (frequency imbalance) were the primary driver, we would expect the top incorrect predictions to consistently skew toward the most frequent classes. However, our analysis of the top-3 incorrect predictions across both object and state in Fig. 3 reveals no such systematic pattern. The absence of this correlation suggests that the observed error stems from the representation failures (internal feature weighting) rather than frequency imbalance.
>
> # Weakness 5
> While we adopt the high-level framework of object and state from prior works like Change-it-Frames, our theoretical contribution fundamentally differs in how we model the interaction between these concepts. Our work provides rigorous examination of this object-state interplay. We move beyond simple categorization to model the conditional dependence that shows how specific objects can support specific states. By testing across 23 models, we show that this failure to model interaction is a systematic gap, *not an artifact of a specific architecture*. As demonstrated by our extensive experimental suite in section 3.3.2, we capture these interactions in ways that other baselines do not.

---

> ### Author Response · Authors · 2025-11-25
> **Response Part 2**
>
> # Question 1
> During our investigation, we looked at potential correlations of bias compared to both model size and release date across all of the tested models. The visualization revealed a high variance in bias that did not align with encoder scale or architecture type and so we did not include this within the paper.
> We also found that there was no difference between VLM-style models and CLIP-style models in our study. We found no empirical justification to isolate and investigate specific encoder mechanics such as the differences between ViT-L and Swin.
>
> # Question 2
> As detailed in section 3.3.2 we conducted specific experiments (Conditioned State/Object) where the textual prompts provided ground truth information for either the object or the state while predicting the other. Our results show that providing this knowledge does slightly improve the model state performance, however, it was still observed that the models were primarily object biased suggesting that providing such knowledge is not enough to solve this problem.
>
> # Question 3
> Our primary research question focuses on the intrinsic availability of state-sensitive information within pre-trained models' representation space. While linear probing is lighter than full fine-tuning, it still necessitates training a classifier. A successful probe proves that information can be extracted with supervision, not necessarily that the model uses this information in its zero-shot setting.  Our qualitative analysis showed that while in some cases distractions in the background affected the model performance, there was no clear pattern to which we can attribute a model making the incorrect prediction.
>
> # Question 4
> Because the data is extracted from YouTube instructional videos, it inherently possesses a wide variety of recording environments. As such, YouTube videos contain different lighting conditions, camera angles and kitchen setups. This randomness means that background noise is likely distributed rather than correlated. The diversity of the "real world" nature of these images reduces the impact of any single lighting condition, making it unlikely that visual confounds alone can explain the consistent performance drop across all experiments.
>
> # Question 5
> While it is plausible that fine-tuning, or training specifically, on such datasets could mitigate object bias, investigating this lies outside the scope of our current study. Our primary objective is to evaluate the intrinsic biases and zero-shot performance of VLMs and we present object bias as a fundamental characteristic emerging from large-scale-pretraining. Training on SSv2 might improve performance on SSv2-like tasks, but it is unclear if this represents genuine reduction in object bias or overfitting to specific state distributions of that dataset. We are interested in whether the models fundamentally understand states, not whether they can be retrained to recognize specific states.

---

### Official Review · Reviewer_vgMx · 2025-11-03

**Soundness:** 2
**Presentation:** 3
**Contribution:** 2
**Rating:** 4
**Confidence:** 4

**Summary:**

The paper aims to analyze the sensitivity of vision-language models (VLMs) to object identity and state. To achieve this, the paper introduces a dataset containing objects in both their original and transformed states. Across a variety of experiments, the paper reveals that VLMs can reliably recognize object categories but struggle to accurately infer states. Moreover, steering models toward greater sensitivity via prompting or injecting oracle information yields marginal improvements. These findings highlight a fundamental limitation in current VLMs. Different training strategies or architectures may be needed to reduce the object-state bias.

**Strengths:**

- The paper introduces a new dataset for analyzing object biases in VLMs.
- Experiments cover a wide range of VLMs with different architectures and different scales, revealing that the object bias is prevalent and exists across almost all models.
- The paper explores mitigation strategies via prompting and reveals that prompt engineering does not fundamentally solve the object bias in VLMs.

**Weaknesses:**

- The proposed dataset is relatively small and restricted to a narrow domain, primarily focusing on kitchen-related objects and activities. This setup limits the dataset’s representativeness and generalizability to broader real-world scenarios.

- The paper does not provide substantial new insights for explaining or addressing bias in VLMs. The discussion attributing object bias to the models’ latent state representations and their training data remains rather general and high-level, without offering concrete empirical evidence.

**Questions:**

Line 193: How do you use a large language model, which does not support images, to do image analysis and select frames?

---

> ### Author Response · Authors · 2025-11-25
> **Response**
>
> # Weakness 1
> The narrow scope of the dataset was an intentional design choice rather than a limitation. Constraining the domain allowed for more precise annotation and more focus on the fine-grained interactions between object and state. The primary motivation for selecting the kitchen domain is that it uniquely features objects that exist in multiple, visually distinct states (e.g. a whole onion vs diced onion). Since objects in the kitchen undergo drastic visual transformation while maintaining their semantic identity this allows us to rigorously test if VLMs can generalize across visually distinct states of the same object as well as the models understanding of an object and its different states. It is also worth noting that the kitchen domain is a heavily researched domain in computer vision with major real-world applications, including assistive robotics, elderly care monitoring, and smart home automation.
>
> # Weakness 2
> While prior literature acknowledges object and state bias tangentially, it has not been subjected to the systematic examination necessary for effective mitigation. We argue that our contribution provides substantial new insight by contextualizing this bias within a rigorous quantitative framework. Unlike previous high-level discussions, we test a wide range of models to provide concrete empirical evidence showing that the object bias is consistent and statistically significant across multiple VLM architecture. Specifically we isolate the relationship between object and state by systematically including and removing their corresponding data in our prompts. These experiments show that the bias persists across varying levels of information which helps in filling a critical gap in the field's ability to diagnose potential causes of failure for real-world applications.
>
> # Question 1
> We used the Multi-modal version of the Llama 3.1-70B model, which accepts both image and text. Given the frame of the video the model is asked to rate how confident it thinks a given object and state are in a given frame. The prompt is provided in Appendix B.

---

### Author Response · Authors · 2025-12-03
**Paper Changes**

A new version was uploaded with changes made in red for easier visibility

---

### Meta-Review · Area_Chair_So35 · 2026-01-05

**Summary:**

The manuscript investigates VLMs' ability to classify objects and their states on a dataset of kitchen related items containing objects in both their original and transformed states (mostly fruit). The analysis reveals that object categories are more easily classified that states.

**Reviewer Concerns:**

The reviews point out diverse concerns regarding
1. The size of the dataset and its narrow focus
2. Limited insights
3. Potential limitations in prompt optimization which could limit the significance of results
4. Lack of Human baselines
5. Lack of disentanglement between  dataset bias (frequency imbalance) and representation bias (model internal weighting).
6. Potential leakage of the analysed bias on the dataset curation itself.
7. Unclear definition of the central term "bias".
and several missing details.
The rebuttal addresses several of these points. In particular, missing details are provided. The authors argue in favor of the narrow scope which enables better interpretability of the results and a controlled scenario. However, the AC is not fully convinced by this argument. It remains unclear, whether the observation (object class is easier than object state for VLMs) is in fact limited to this narrow scope.
Other point like the human study and the alternatives in prompting are described in the rebuttal text but are missing in the revised document so that the provided details are not sufficient.

**Reviewer Scores:**

The reviewers have provided the following scores:
Reviewer vgMx: 4 --> I don't think the authors arguments in favor of a narrow scope dataset are convincing. The concern of a too small contribution would likely persist.
Reviewer UCUV: 4 --> could have increased rating to 6 after rebuttal.
Reviewer qmb8: 4 --> the concerns are not properly addressed in the revision.
Reviewer x27o: 0 --> the questions are addressed but the concerns are not addressed. The reviewer would likely not increase the score towards acceptance.
In summary, after the rebuttal, some concerns and minor questions by reviewers have been addressed but the most important points are still valid.

---

### Decision · Program_Chairs · 2026-01-26

Reject